# Model-Based Estimates for Farm Labor Quantities

**Lu Chen** [1,2,*] , **Nathan B. Cruze** [3] **and Linda J. Young** [2]

1   National Institute of Statistical Sciences, 1750 K Street NW Suite 1100, Washington, DC 20006, USA
2   United States Department of Agriculture, National Agricultural Statistics Service, 1400 Independence Avenue SW, Washington, DC 20250, USA; linda.j.young@usda.gov
3   NASA Langley Research Center, Mail Stop 290, Hampton, VA 23681, USA; nathan.b.cruze@nasa.gov
*   Correspondence: lchen@niss.org

**Abstract:** The United States Department of Agriculture's (USDA's) National Agricultural Statistics Service (NASS) conducts the Farm Labor Survey to produce estimates of the number of workers, duration of the workweek, and wage rates for all agricultural workers. Traditionally, expert opinion is used to integrate auxiliary information, such as the previous year's estimates, with the survey's direct estimates. Alternatively, implementing small area models for integrating survey estimates with additional sources of information provides more reliable official estimates and valid measures of uncertainty for each type of estimate. In this paper, several hierarchical Bayesian subarea-level models are developed in support of different estimates of interest in the Farm Labor Survey. A 2020 case study illustrates the improvement of the direct survey estimates for areas with small sample sizes by using auxiliary information and borrowing information across areas and subareas. The resulting framework provides a complete set of coherent estimates for all required geographic levels. These methods were incorporated into the official *Farm Labor* publication for the first time in 2020.

**Keywords:** agricultural survey; auxiliary data; Bayesian diagnostic; official statistics; small area estimation; subarea models





## 1. Introduction

The United States Department of Agriculture's (USDA's) National Agricultural Statistics Service (NASS) conducts the Farm Labor Survey, which provides the basis for employment and wage estimates for all workers directly hired by farms and ranches in all states (excluding Alaska). The NASS Farm Labor Survey is conducted semi-annually. Each collection captures data for reference weeks in two distinct quarters. The April survey acquires data for the January and April quarters, and the October survey collects data with reference to July and October. The resulting official statistics provide important estimates of demand for seasonal agricultural workers, which are used by other federal and state agencies for decision making and research. The Department of Labor references annual average wage rates for some published domains when setting statutory minimum wage payments for immigrant farm workers under the H-2A visa program.

In the traditional process of setting official statistics, the NASS Agricultural Statistics Board uses expert opinion to synthesize information from the current year's direct survey estimates, the prior year's published statistics, and input from statisticians in NASS regional field offices. The use of expert opinion did not lead to valid measures of uncertainty and resulted in a lack of transparency and reproducibility. The motivation of the Farm Labor project is to implement small area models for integrating survey data with multiple sources of information to provide more precise official estimates with measures of uncertainty. Here, we discuss how to provide a complete set of reliable and coherent estimates for all required geographic levels and worker types of interest. These methods were successfully incorporated into the official *Farm Labor* publication for the first time in 2020 (see USDA NASS [1–3] for the full contents of these reports).

In recent years, small area models have drawn the attention of academic researchers and national statistical offices. Small area estimation models can "borrow strength" from related areas across space and time or through auxiliary information that is correlated with the variable of interest to provide "indirect" estimates with increased precision for small areas. In its *Farm Labor* publication, NASS publishes tables of estimates for various worker classes at the regional and US level by aggregating state-level estimates. Some of these state-level cells may be based on few reports of items of interest, resulting in issues of data quality or disclosure limitations. From a data quality perspective, only using the direct survey estimates based on a small sample size may not be precise or may not be possible (Rao and Molina [4]). However, model-based estimates incorporating direct estimates and auxiliary information can potentially improve the precision and lead to improved reproducibility and transparency. To adopt a model-based approach for the farm labor program, the estimates must maintain harmony among nested levels while providing measures of uncertainty. It is also desired, when possible, that the current year's survey estimates and the current year's official statistics reflect the same direction of change from the previous year's official statistics.

Two major types of small area models, area-level and unit-level models, have been developed based on both frequentist and Bayesian methods. Pfeffermann [5] and Rao and Molina [4] provided a comprehensive overview of the development, methods, and application of small-area estimation, including various types of area-level and unit-level models. Walker et al. [6] compared the Huddleston–Ray and Battese–Fuller estimators for incorporating survey data and satellite data to improve acreage estimates. Subsequently, the celebrated work of Battese et al. [7] introduced the unit-level models for small area estimation based on nested error linear regression using combinations of survey and satellite data. Since the computation time of unit-level models is frequently much longer than it is for area-level models, which might not be feasible in production, the focus for this application is on area-level models. For continuous responses, the first and most common model is the Fay-Herriot model (Fay and Herriot [8]) in small area estimation. It is an area-level model based on a "normal-normal-linear" assumption. In many small area applications, the Fay-Herriot model is fitted on a transformed scale. For example, when data are nonlinear on the original scale or highly skewed to the right, the log-transformed Fay-Herriot model is frequently employed. In this paper, subarea-level models, as extensions of area-level models, are developed. The distribution of the data is carefully checked to ensure the models' assumptions are met.

Recent studies and papers related to NASS small area estimation research on county estimates of crops have shown that the hierarchical Bayesian small area models can incorporate auxiliary sources of data with survey estimates to improve the precision and increase the accuracy of related NASS official estimates. Erciulescu et al. [9] explored preservation of the triplet relationships among numerator totals, denominator totals, and their ratios for two nested, smaller-than-state geographies. The authors developed Bayesian subarea-level models for two of the three quantities (either the denominator total and numerator total or the denominator total and ratio) and used samples from posterior distributions to derive distributions and Monte Carlo estimates of the third quantity subject to external state-level benchmarks. Erciulescu et al. [10] proposed and implemented a double shrinkage hierarchical Bayesian subarea-level model to provide the acreage estimates with associated measures of uncertainty. After integrating different data sources, the county-level model-based acreage estimates decreased the coefficients of variance relative to those of the direct estimates. Erciulescu et al. [11] discussed the challenges of missing data, either survey responses or administrative data, when fitting the hierarchical Bayesian subarea-level model to obtain the total crop estimates for the whole nation. Chen et al. [12] implemented hierarchical Bayesian subarea-level models with inequality constraints to produce county-level estimates that satisfy important relationships between the estimates and administrative data, along with the associated measures of uncertainty.

Adding to these recent efforts at NASS, models for three different estimates of interests are discussed and applied to the first quarter of the 2020 Farm Labor data. In Section 2, a description of the Farm Labor Survey is presented. In Section 3, an exploratory data analysis of the quantities of interest is presented, and two hierarchical Bayesian subarea models are developed. In Section 4, the construction of internally benchmarked, aggregated posterior summaries for the totals and ratios at the draw level—instead of being applied only to the point estimates—is discussed. In Section 5, a case study based on the January 2020 data highlights the performance of the model-based estimates vis-a-vis the survey estimates. Conclusions and future work are presented in Section 6.

## 2. Farm Labor Survey

The Farm Labor Survey is conducted semi-annually in April and October in all surveyed states except California. Data for the reference weeks in January and April are collected in the April survey. During the October data collection, data for both the July and October reference weeks are collected. For California, the collection of these data is currently conducted on a quarterly basis in collaboration with California's Employment Development Department. See the Farm Labor Methodology and Quality Measures document [3], which is provided with each semi-annual data release, for more detail on the NASS Farm Labor Survey.

The target population for the Farm Labor Survey program is any farm or ranch with USD 1000 or more in annual agricultural sales (or potential sales). The Farm Labor Survey is a multiple frame survey, incorporating samples from both the NASS list and area frames to ensure adequate coverage. Farms on the list frame are sampled through a hierarchical stratified design, with strata based on measures of size. In order to provide coverage of farms not currently part of the NASS list frame, a second sample is drawn consisting of segments of land selected from the NASS area frame. Throughout the data collection process, steps are taken to minimize the impact of nonsampling errors such as reporting, recording, and editing errors.

The collected data are weighted according to the survey design and adjusted for nonresponse, and the direct estimates and estimated sampling variances of the number of hired workers, average hours worked per week, and wage rates are summarized at the state, regional, and national levels. The phrase "NASS worker types" is used to draw a distinction between one publication's categorization of workers (based on four fundamental categories: field, livestock, supervisory, and 'other' types, as well as the aggregates thereof) and a finer detail of categorization provided by the Standard Occupational Classification (SOC) system. Direct estimates are available for states, regions, and the nation, which are crossed with the NASS worker type, type of farm (e.g., field crops, other crops, livestock, and poultry operations), economic class of the farm (binned into six categories by gross value of sales), or SOC code. In addition, estimates for combined worker types at the regional or US levels are also required. For example, at the regional and US levels, NASS publishes the wage rates for all hired workers by averaging the four fundamental NASS worker types, and the wage rates are averaged over a combination of the field and livestock worker types. The latter estimate has importance in the administration of minimum wages paid to immigrant farm workers under the H-2A visa program.

All the estimates are published in the semi-annual *Farm Labor* release, whcih is issued around the 21st of May or November. Apart from California, Florida, and Hawaii, which publish region-level equivalents, farm labor regions consist of groups of two or more neighboring states, as shown in Figure 1. Both semi-annual releases include regional and US-level estimates for the number of workers, average hours worked, and average wage rates by type of worker. The November release also includes annual average estimates at the regional and US levels for both NASS worker types and SOC categories. The annual average wage rate for NASS field and livestock workers (combined) is referenced specifically in the Code of Federal Regulations in defining the Adverse Effect Wage Rate (See *CFR* Title 20, Chapter V, Part 655, Subpart B, https://www.ecfr.gov/current/title-20/chapter-V/part-65

, accessed on 12 February 2010), a statutory measure that may determine the minimum wage that can be paid to H-2A workers. These NASS estimates are used by the Department of Labor to fulfill its statutory obligations and to provide transparency of the manner in which minimum wages are calculated.

In this paper, small area models are implemented that provide coherent model-based estimates of the number of hired workers, average hours worked, and average wage rate. For brevity, the focus is on estimates for NASS worker types for each quarter at required geographic levels. These estimates make up the bulk of the NASS *Farm Labor* publication and fulfill the agency's obligations to the Department of Labor.

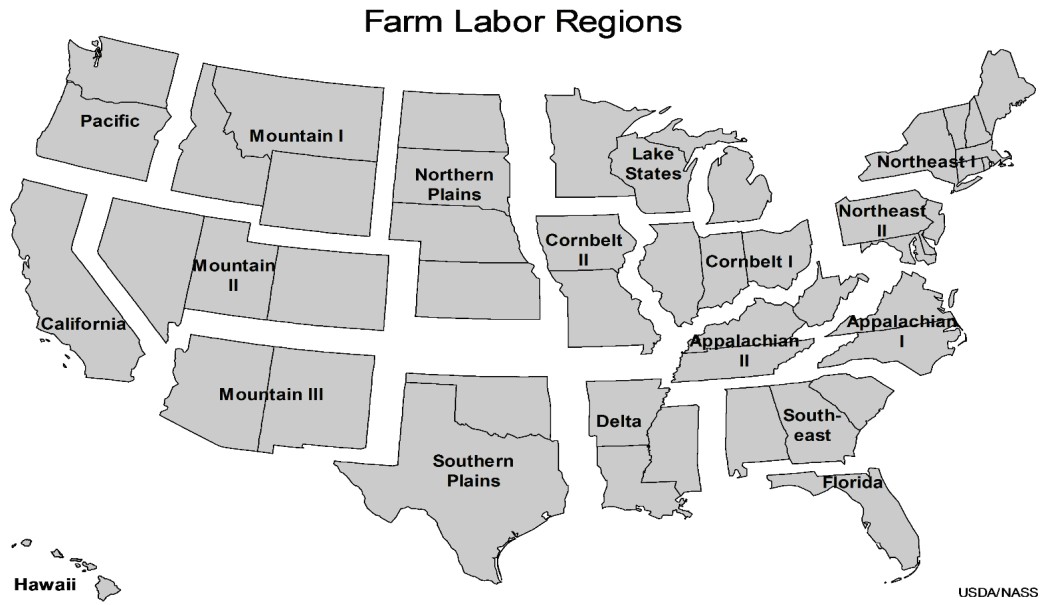

**Figure 1.** The 18 farm labor regions. Three states (California, Florida, and Hawaii) publish as their own regions.

### 3. Models

Several hierarchical Bayesian subarea-level models are discussed for estimating different quantities of interest: the number of workers, hours worked per week, and average wage rates based on NASS worker type tables. The area represents the region, and the subarea represents the state within a region. The smallest domain is stated by each of the four fundamental NASS worker types. Exploratory data analysis informs the choice of different sampling models for worker totals versus other quantities of interest.

One advantage of Bayesian methods is that they provide a straightforward framework for constructing summaries for the model estimates, including posterior means and posterior variances. The posterior distributions obtained from Bayesian models can provide a rich output that can be used to construct aggregated posterior summaries for any aggregate of states or combination of worker types. As mentioned before, state-level summaries based on record-level responses are the lowest level of geography for which Farm Labor Survey direct expansions are formed. However, NASS also publishes estimates at the regional and US levels, and other estimates are produced for combined worker types. For example, one table shows the quarterly number of all hired workers for each region in the US. All hired workers include all worker types: field workers, livestock workers, supervisors, and other workers. The aggregated regional estimates are the estimated numbers of hired workers for all states within each region. A second tabulation provides the quarterly wage rates of the following types of workers: field workers, livestock workers, field and livestock workers (combined), and all hired workers. Since wage rates are defined as the ratio of total wages to total hours worked, the aggregated wage rate estimates are the weighted averages of the total hours worked based on the number of workers and hours per week (per worker)

estimates. These two examples of the numerous estimates published for cross-tabulations of aggregated domains illustrate that point predictions at both the subarea and area levels are of interest and help motivate the choice of Bayesian methods for NASS applications.

### 3.1. Exploratory Data Analysis

Before fitting the models, an exploratory data analysis is conducted to evaluate the distributions of the survey estimates. For the number of workers, the estimates are naturally non-negative, and their distribution is highly skewed to the right. (See Figure 2a for a density plot of the January 2020 survey estimates for workers totals, which includes all four fundamental worker types.) Some states and worker types are heavily impacted the totals. Log-transformation can make the distribution of transformed estimates more symmetric and promise non-negativity (see Figure 2b). Trevisani and Torelli [13] discussed several hierarchical Bayesian small area models for counting data. One of the models with log-transformed data is the hierarchical Bayesian log-normal-normal (LNN) model. It is a matched model [14] in the sense that sampling and linking models can be combined to produce a linear mixed model. The model based on the log transformation had better results when compared with the original normal model. Log transformation may be sensible for additive and linear small area models. The county-level model of the counts of poor people in the US Census Bureau's Small Area Income and Poverty Estimates (SAIPE) program is based on log-transformed data. An overview of the methodology for the SAIPE program can be found in the work of Bell et al. [15].

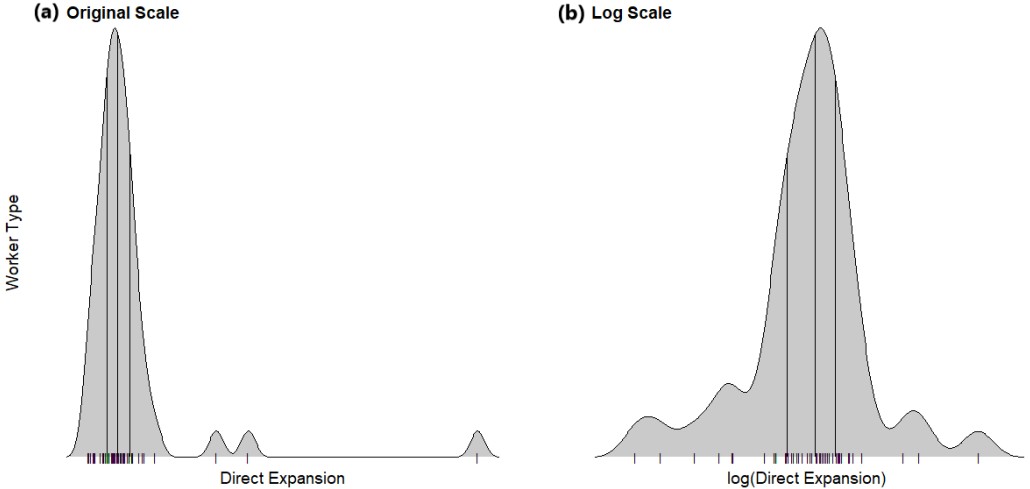

**Figure 2.** (**a**) Density plot of the original state-level survey estimates of number of workers, January 2020. (**b**) Density plot of the log-transformed survey estimates of number of workers.

Although estimates of hours and wage rates must also be non-negative, the distributions of state-level direct estimates of the average hours per week and wage rates by worker type were approximately symmetrical (see Figures 3 and 4, respectively). The horizontal axes were redacted to avoid disclosing magnitudes in unpublished survey categories or geographies. However, the horizontal ranges were controlled in each figure, facilitating comparison of the location and spread in the distributions of the observed direct estimates across worker types. Note the larger typical wage in the supervisory category, as well as the increased range in supervisory and other worker types relative to either the crop or livestock workers (see Figure 4). Therefore, normal subarea-level models ([4,11,16]) can be applied to both variables of interest.

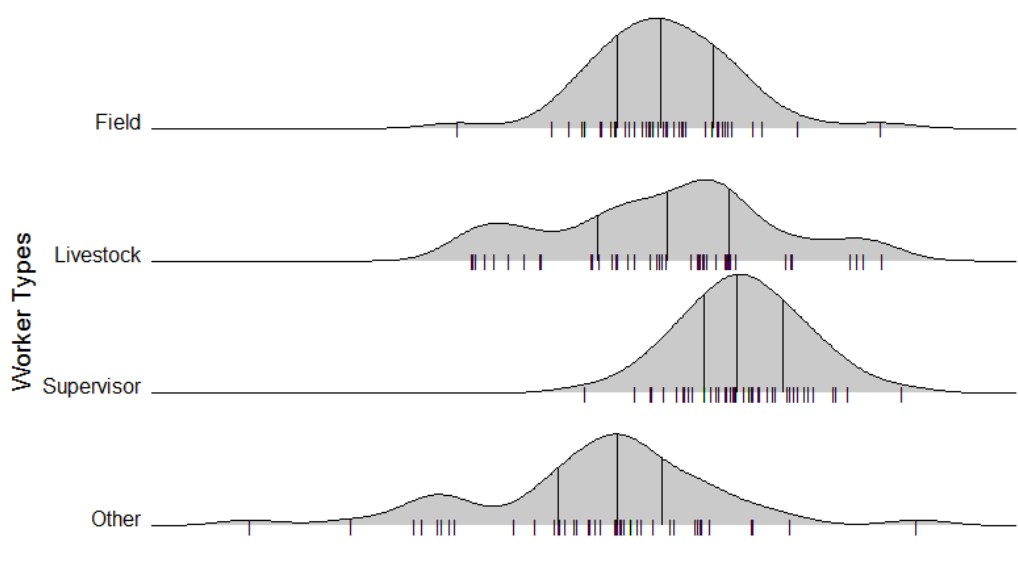

**Figure 3.** Density plots of the original January 2020 state-level survey estimates of hours per week by each worker type: field workers, livestock workers, supervisors, and other types of workers.

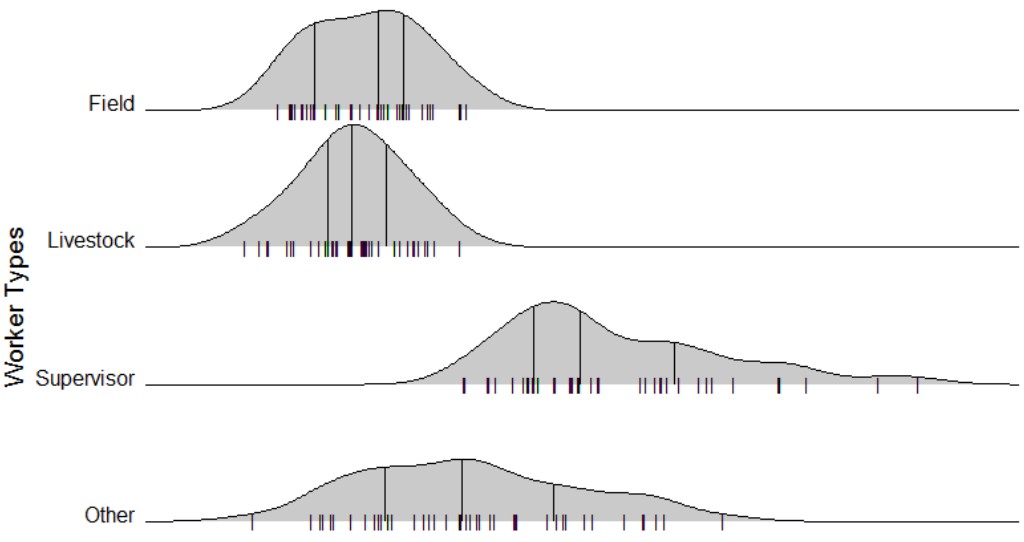

**Figure 4.** Density plots of the original January 2020 state-level survey estimates of wage rates (USD/hour) by worker type: field workers, livestock workers, supervisors, and other types of workers.

### 3.2. Log-Normal-Normal Subarea Model

Based on the exploratory analysis of the number of workers discussed in Section 3.1, a hierarchical Bayesian LNN subarea model was implemented. In the case study for 2020, no zeros or missing direct expansions for any NASS worker type occurred at the state level. The observations were the log-transformed, state-level direct expansions of the number of workers and associated known sampling variances.

Let $i \in \{1, 2, \ldots, 18\}$ be an index for 18 Farm Labor regions and $j = 1, \ldots, n_i$ be an index on states in the $i$th region. Let $k \in \{1, 2, 3, 4\}$ be an index for the fundamental NASS worker types. The survey's direct estimate of the number of workers for type $k$ in state $j$

and region $i$ is denoted by $\hat{y}_{ijk}$, and the associated survey variance is $\hat{\sigma}_{ijk}^2$. $Y_{ijk}$ denotes the true population number of workers that $\hat{y}_{ijk}$ is intended to estimate. The auxiliary data used in the models are $x_{ijk}$, including an intercept. The notation $y|x$ is used to denote the conditional distribution of $y$ given $x$.

The LNN subarea model is

$$\hat{y}_{ijk}^* = \log(\hat{y}_{ijk})|\theta_{ijk} \overset{ind}{\sim} N(\theta_{ijk}, \hat{\sigma}_{ijk}^{*2}), \ k = 1, \dots, 4,$$

$$\theta_{ijk}|\boldsymbol{\beta}, \nu_i, \sigma_\mu^2 \overset{ind}{\sim} N(\mathbf{x}_{ijk}'\boldsymbol{\beta} + \nu_i, \sigma_\mu^2), j = 1, \dots, n_i,$$

$$\nu_i|\sigma_\nu^2 \overset{iid}{\sim} N(0, \sigma_\nu^2), \ i = 1, \dots, 18, \tag{1}$$

$$\boldsymbol{\beta} \sim MN(\hat{\boldsymbol{\beta}}, 1000 \times \hat{\Sigma}_{\hat{\boldsymbol{\beta}}}),$$

$$\sigma_\mu^2 \sim \text{Uniform}(R^+), \ \sigma_\nu^2 \sim \text{Uniform}(R^+),$$

where, following the Delta method, $\hat{\sigma}_{ijk}^{*2} = (\hat{y}_{ijk})^{-2}\hat{\sigma}_{ijk,y}^2$ serves as an estimate for the sampling variances after log transformation. The term $\nu_i$ is the area-level random effect representing the region-level variability. An empirical diffuse prior is assigned to the coefficients $\boldsymbol{\beta}$ (i.e., a multivariate normal prior distribution with a fixed and known mean and variance and a covariance matrix $\boldsymbol{\beta} \sim MN(\hat{\boldsymbol{\beta}}, 1000\hat{\Sigma}_{\hat{\boldsymbol{\beta}}})$). Here, $\hat{\boldsymbol{\beta}}$ are the least squares estimates of $\boldsymbol{\beta}$ obtained from fitting a simple linear regression model of the county-level survey estimates on the auxiliary data $x_{ij}$, and $\hat{\Sigma}_{\hat{\boldsymbol{\beta}}}$ is the estimated covariance matrix of $\hat{\boldsymbol{\beta}}$. The prior distributions for $\sigma_\mu^2$ and $\sigma_\nu^2$ are noninformative, uniform$(R^+)$, where $R^+$ represents the positive real numbers. The discussion in Browne and Draper [17] motivates the use of a uniform prior distribution for the random-effect variance components.

### 3.3. Normal-Normal Subarea Model

Based on exploratory analysis of the average hours worked per week and average wage rate per hour discussed in Section 3.1, the hierarchical Bayesian normal-normal (NN) subarea model was applied to both quantities. When modeling these two quantities of interest, the observations were the state-level survey ratio estimates and the associated known sampling variances on the original scale. Torabi and Rao [16] illustrated the most frequent approaches to model fitting and estimation for the subarea-level model with known sampling variances. Erciulescu et al. [11] proposed and discussed the subarea-level model using a Bayesian approach. Because the interest is in constructing summaries for all different levels, a Bayesian approach to model fitting and estimation is preferable.

In the model specified below for the ratio variables, indexes $i$, $j$, and $k$ retain the same meaning. The direct survey estimates of either the average hours per week or the average wage rate per hour for type $k$ in state $j$ and region $i$ is denoted by $\hat{y}_{ijk}$, and the associated survey variance is $\hat{\sigma}_{ijk}^2$. $Y_{ijk}$ is denoted as the true population value of the number of workers that $\hat{y}_{ijk}$ is intended to estimate. The auxiliary data used in the models are $x_{ijk}$, including an intercept.

The NN subarea model is

$$\hat{y}_{ijk}|\theta_{ijk} \overset{ind}{\sim} N(\theta_{ijk}, \hat{\sigma}_{ijk}^2), \ k = 1, \dots, 4,$$

$$\theta_{ijk}|\boldsymbol{\beta}, \nu_i, \sigma_\mu^2 \overset{ind}{\sim} N(\mathbf{x}_{ijk}'\boldsymbol{\beta} + \nu_i, \sigma_\mu^2), j = 1, \dots, n_i,$$

$$\nu_i|\sigma_\nu^2 \overset{iid}{\sim} N(0, \sigma_\nu^2), \ i = 1, \dots, 18, \tag{2}$$

$$\boldsymbol{\beta} \sim MN(\hat{\boldsymbol{\beta}}, 1000 \times \hat{\Sigma}_{\hat{\boldsymbol{\beta}}}),$$

$$\sigma_\mu^2 \sim \text{Uniform}(R^+), \ \sigma_\nu^2 \sim \text{Uniform}(R^+).$$

Again the term $\nu_i$ is the area-level random effect representing the region-level variability. As before, an empirical diffuse prior is adopted on the coefficients $\boldsymbol{\beta}$, and the prior distributions for $\sigma_\mu^2$ and $\sigma_\nu^2$ are noninformative uniform priors.

The interest in fitting the model is to obtain the posterior distribution of $\theta_{ijk}$. After model fitting, the identity transformation defines the estimators $Y_{ijk}^S = \theta_{ijk}$ for $S \in \{hr, wg\}$ that can be applied to obtain the posterior means and measures of uncertainty for the average hours worked per week (hr) and the average wage per hour (wg) of each worker type. Conditional on the state-level MCMC samples of the number of workers, aggregated region-level posterior summaries for the hours and wage rates of different worker types can be obtained. Details are shown in Section 4.

## 4. Aggregating Posterior Samples

In the NASS publication, the regional and national level official estimates are released in May and November. NASS headquarters completes the regional and national level summaries, which utilize the state-level summaries. Estimates for larger geographies typically have been calculated directly from state level sub-components; that is, in effect, NASS has had a tradition of internally benchmarking this data product. In a similar manner, after the modeling for subarea estimates in each quarter, the Monte Carlo samples can be used to derive entire distributions for aggregated estimates by operations on the posterior samples obtained for the subareas. These aggregated results also provide measures of uncertainty. The posterior distribution of all aggregated quantities of interest can be obtained as well. In this section, the iteration method for estimating the aggregated summaries, which involve higher geographical levels and combined worker types, is introduced.

Let $Y_{ijk}^S$, $S \in \{wk, hr, wg\}$ denote the true parameters of interest for the $k$th worker type in the $j$th subarea and the $i$th area, corresponding to the number of workers, average hours per week, and average wage rate per hour, respectively. Let $y_{ijk}^{S,(h)}$ denote the draws of $Y_{ijk}^S$, where $h$ indexes the draws from the posterior distribution and $h = 1, \dots, H$. In our application, the smallest domain for the model estimates is at the state level in each worker type.

As seen during the exploratory data analysis, the number of workers, which is the response variable, had a distribution highly skewed to the right. However, the distribution of the logarithm of the number of workers (the response) was approximately symmetrical. Thus, the LNN model in Equation (1) was adopted. We used the Bayesian way [13] to transform $\theta_{ijk}$ back to the original scale, providing $\hat{Y}_{ijk}^{wk}$ as posterior expectation of the back-transformed parameter $Y_{ijk}^{wk} = \exp(\theta_{ijk})$. This is essentially the median of the distribution in the original scale. Given the skewness of the lognormal distribution, the median provides a better measure of the middle of the distribution (Goyal et al. [18] and Rao and D'Cunha [19]).

The number of workers parameter is a total. Summation of the corresponding iterates with respect to the worker type or geography results in a Monte Carlo sample for the aggregated total estimates. If only considering a higher geographic level, the regional estimate of the number of workers for each type at the draw level is $y_{ik}^{wk,(h)} = \sum_{j=1}^{n_i} y_{ijk}^{wk,(h)}$, and the national estimate of the number of workers is $y_k^{wk,(h)} = \sum_{i=1}^{18} \sum_{j=1}^{n_i} y_{ijk}^{wk,(h)}$. When aggregating by both the geographic level and worker type, for example, the national level estimate of the number of all hired workers at the draw level is $y^{wk,(h)} = \sum_{k=1}^{4} \sum_{i=1}^{18} \sum_{j=1}^{n_i} y_{ijk}^{wk,(h)}$, $h = 1, \dots, H$. Therefore, it is straightforward to construct the posterior means and posterior variances based on draws at different aggregated levels. We continue with the national level estimate of the number of all hired workers as the example. The posterior mean is estimated by

$$\hat{E}(Y^{wk}|data) = H^{-1} \sum_{h=1}^{H} y^{wk,(h)} = \tilde{y}^{wk}. \qquad (3)$$

The posterior variance is estimated by

$$\hat{Var}(Y^{wk}|data) = (H-1)^{-1} \sum_{h=1}^{H} \left( y^{wk,(h)} - H^{-1} \sum_{h=1}^{H} y^{wk,(h)} \right)^2. \tag{4}$$

The average hours per week (per worker) and the dollar per hour wage rate parameters ($Y_{ijk}^{hr}$ and $Y_{ijk}^{wg}$, respectively) are ratios. Hours are calculated as the ratio of the total number of hours worked to the total workers employed in the reference week. The wage rates are calculated as the ratio of total wages paid to total hours worked. We can construct each of these totals across the required geographic and worker type levels to form the estimates at the draw level, similar to the method for the number of worker estimates. If only considering a higher geographic level, for each reference week, the regional level estimate of total hours worked for each type at the draw level is $\sum_{j=1}^{n_i} y_{ijk}^{wk,(h)} y_{ijk}^{hr,(h)}$, and the regional level estimate of total wages for each worker type at the draw level is $\sum_{j=1}^{n_i} y_{ijk}^{wk,(h)} y_{ijk}^{hr,(h)} y_{ijk}^{wg,(h)}$. Then the regional level estimate of average hours for each worker type at draw level is

$$y_{ik}^{hr,(h)} = \frac{\sum_{j=1}^{n_i} y_{ijk}^{wk,(h)} y_{ijk}^{hr,(h)}}{\sum_{i=1}^{18} \sum_{j=1}^{n_i} y_{ijk}^{wk,(h)}}, \tag{5}$$

and the corresponding wage rate for each worker type is

$$y_{ik}^{wg,(h)} = \frac{\sum_{j=1}^{n_i} y_{ijk}^{wk,(h)} y_{ijk}^{hr,(h)} y_{ijk}^{wg,(h)}}{\sum_{i=1}^{18} \sum_{j=1}^{n_i} y_{ijk}^{wk,(h)} y_{ijk}^{hr,(h)}}, \tag{6}$$

where $i = 1, \ldots, 18$ and $h = 1, \ldots, H$.

As an example of aggregating at both the geographic level and by worker type, the national level estimate of the total hours and total wage rates for all hired workers at the draw level are $\sum_{k=1}^{4} \sum_{i=1}^{18} \sum_{j=1}^{n_i} y_{ijk}^{wk,(h)} y_{ijk}^{hr,(h)}$ and $\sum_{k=1}^{4} \sum_{i=1}^{18} \sum_{j=1}^{n_i} y_{ijk}^{wk,(h)} y_{ijk}^{hr,(h)} y_{ijk}^{wg,(h)}$, $h = 1, \ldots, H$, respectively. Then, the regional level estimate of the average hours for all hired workers at the draw level is

$$y^{hr,(h)} = \frac{\sum_{k=1}^{4} \sum_{i=1}^{18} \sum_{j=1}^{n_i} y_{ijk}^{wk,(h)} y_{ijk}^{hr,(h)}}{\sum_{k=1}^{4} \sum_{i=1}^{18} \sum_{j=1}^{n_i} y_{ijk}^{wk,(h)}}, \tag{7}$$

and the corresponding wage rate for all hired workers is

$$y^{wg,(h)} = \frac{\sum_{k=1}^{4} \sum_{i=1}^{18} \sum_{j=1}^{n_i} y_{ijk}^{wk,(h)} y_{ijk}^{hr,(h)} y_{ijk}^{wg,(h)}}{\sum_{k=1}^{4} \sum_{i=1}^{18} \sum_{j=1}^{n_i} y_{ijk}^{wk,(h)} y_{ijk}^{hr,(h)}}, \tag{8}$$

where $h = 1, \ldots, H$.

Therefore, we can construct the posterior means and posterior variances based on draws at different aggregated levels. As another illustration, for the national level estimate of the average hours and average wage rates as the example, the posterior means are estimated by

$$\hat{E}(Y^{hr}|data) = H^{-1} \sum_{h=1}^{H} \frac{\sum_{k=1}^{4} \sum_{i=1}^{18} \sum_{j=1}^{n_i} y_{ijk}^{wk,(h)} y_{ijk}^{hr,(h)}}{\sum_{k=1}^{4} \sum_{i=1}^{18} \sum_{j=1}^{n_i} y_{ijk}^{wk,(h)}} = \tilde{y}^{hr}, \tag{9}$$

$$\hat{E}(Y^{wg}|data) = H^{-1} \sum_{h=1}^{H} \frac{\sum_{k=1}^{4} \sum_{i=1}^{18} \sum_{j=1}^{n_i} y_{ijk}^{wk,(h)} y_{ijk}^{hr,(h)} y_{ijk}^{wg,(h)}}{\sum_{k=1}^{4} \sum_{i=1}^{18} \sum_{j=1}^{n_i} y_{ijk}^{wk,(h)} y_{ijk}^{hr,(h)}} = \tilde{y}^{wg}. \tag{10}$$

The posterior variance is estimated by

$$\hat{Var}(Y^{hr}|data) = (H-1)^{-1} \sum_{h=1}^{H} \left(y^{hr,(h)} - \tilde{y}^{hr}\right)^2, \tag{11}$$

$$\hat{Var}(Y^{wg}|data) = H^{-1} \sum_{h=1}^{H} \left(y^{wg,(h)} - \tilde{y}^{wg}\right)^2. \tag{12}$$

These draw-level estimators and estimators of the posterior means and variances can be similarly applied to obtain the corresponding estimator for the statutory target estimators (crop and livestock workers combined) by writing, without loss of generality, summation over $k \in \{1,2\}$ instead of $k \in \{1,2,3,4\}$.

## 5. Case Study

### 5.1. Model Fit and Estimation

The two models introduced in Section 3 are useful tools for producing model-based estimates of NASS worker types from Farm Labor data. In this section, data with reference weeks in January 2020 were selected as the case study because it was one of the quarters NASS incorporated these methods to produce official statistics. The LNN subarea model was applied to all states with positive data to produce model-based estimates and summaries of the number of workers for each worker type. The NN subarea model was applied to all states with positive data to produce model-based estimates and summaries of the average hours and average wage rates for each worker type, respectively. There are 44 states (By convention, data for New Hampshire, Rhode Island, Connecticut, Massachusetts, Vermont, and Maine are processed together as a single state, subdividing the Northeast I region as New York and all other New England states.) within 18 regions for 4 different worker types.

The covariate $\mathbf{x}_{ijk}^{wk}$, used in the LNN model for the number of workers, includes an intercept, the log of the same corresponding quarter, the previous year's published estimates, the positive number of reports, and the indicator variables encoding the four worker types. In this case, $k = 1, 2, 3, 4$ corresponds to the worker types of field workers, livestock workers, supervisors, and other workers, respectively. The covariate $\mathbf{x}_{ijk}^{hr}$, used in the NN model for the average hours, includes an intercept, the same corresponding quarter, the previous year's published estimates, the positive number of reports, and indicator variables encoding the four worker types. The covariate $\mathbf{x}_{ijk}^{wg}$, used in the NN model for wage rates, is very similar to the one in the model for estimates of the average hours. As mentioned before, the relationship between the current year's survey estimates and the previous year's official values is one of the key factors considered in the Agricultural Statistics Board process. Therefore, the same-quarter previous year's published estimates are included in the models.

Convergence diagnostics were conducted. The convergence was monitored using trace plots, the multiple potential scale reduction factors ($\hat{R}$ close to 1), and the Geweke test of stationarity for each chain (Gelman and Rubin [20] and Geweke [21]). Additionally, once the simulated chains mixed, the effective number of independent simulation draws to monitor simulation accuracy was determined. Computation time is an additional factor when candidate models are evaluated for use in production. The described models and their posterior summaries, which generate estimates and measures of uncertainty for thousands of cells per quarter for each of two quarters, must be completed in less than 1 day during the semi-annual production windows.

### 5.2. Results

The state-level model-based estimate (ME) of $Y_{ijk}^S$ is denoted by $\tilde{y}_{ijk}^S$ and computed as the posterior mean of $Y_{ijk}^S$. The required aggregated estimates and corresponding variances can be obtained as discussed in the Section 4.

The state-level, model-based estimates for all hired workers (the aggregate of all four fundamental worker types) $\tilde{y}_{ij}^S$ for $S \in \{wk, hr, wg\}$ were compared with the survey's direct estimates (DE). To avoid disclosing unpublished state-level estimates, the absolute relative differences (ARD) between the survey and model estimates were computed as follows:

$$\text{ARD}(\%) = 100 \times \left| \frac{\text{DE} - \text{ME}}{\text{DE}} \right|.$$

The choropleth map depicts the ARD for the state-level estimate of the total number of all hired workers (see Figure 5). As the difference between the survey estimates and model-based estimates increases (relative to the survey estimate), the corresponding area becomes darker. The model-based estimates were "far" from the direct estimates in areas with small sample sizes but "close" in those with larger sample sizes. Nevada and Utah, shown in dark blue, had small sample sizes when compared with other states. In the areas with smaller sample sizes, the subarea model incorporated other data and also "borrowed information" across and within areas and subareas.

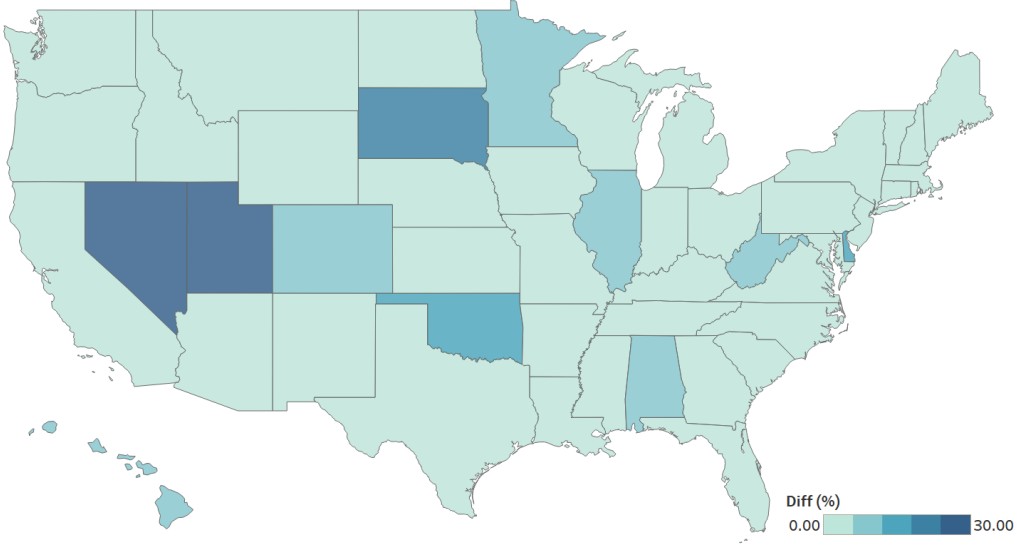

**Figure 5.** Absolute relative differences (%) between state-level survey and model point estimates for estimates of the number of all hired agricultural workers.

The pattern of the ARD for the state-level average wage rate and hours per week estimates, averaged across all four worker types (Figures 6 and 7, respectively), was different from the state-level estimates of the total number of all hired workers (Figure 5). In addition, note the difference in range on the color bars. In addition to small sample sizes, the modeled state estimates of the average wage rates and hours differed substantially from the direct estimates when the difference in the current year's survey estimates and the previous year's published estimates was large. The subarea model provided smoothed estimates by incorporating the previous year's published values.

Another way of illustrating the differences between the model-based estimates and survey's direct estimates is to use the ratios of the model-based estimate to the corresponding survey's direct estimate as a function of the sample size. The ratios can provide further insights into the differences in the two methods of estimation (see Figure 8 for plots of the ratios of ME to DE for the estimated number of workers, hours worked per week, and average wage rates at the state level against the sample sizes). The widest range of ratios between the model estimates and the survey's direct estimates was for states with small sample sizes, and the ratios tended to become closer to one as the sample size increased for all three estimates. This illustrates the shrinkage of the direct estimates toward the modeled (regression) estimates obtained by using all available sources of information.

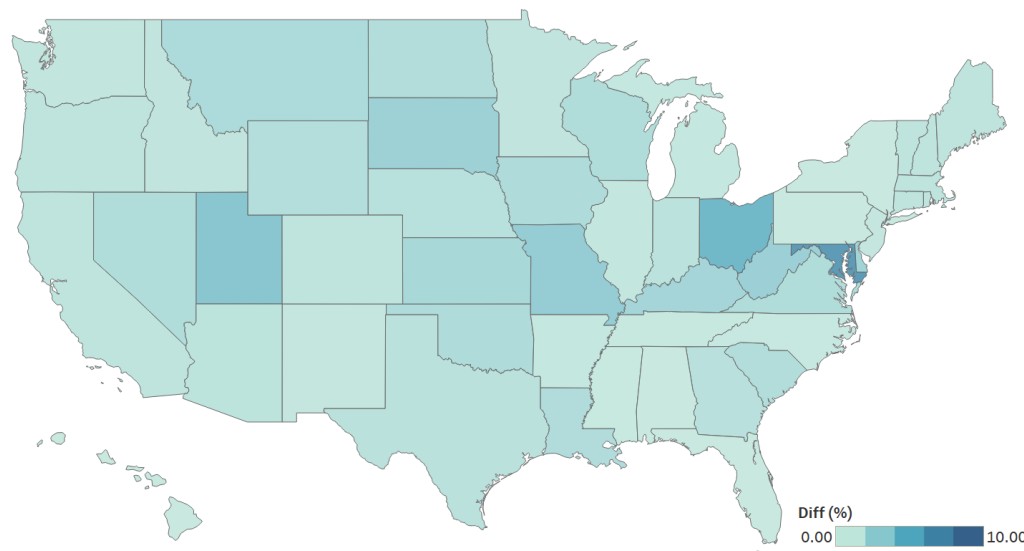

**Figure 6.** Absolute relative differences (%) between state-level survey and model point estimates for the average wage rates of all hired agricultural workers.

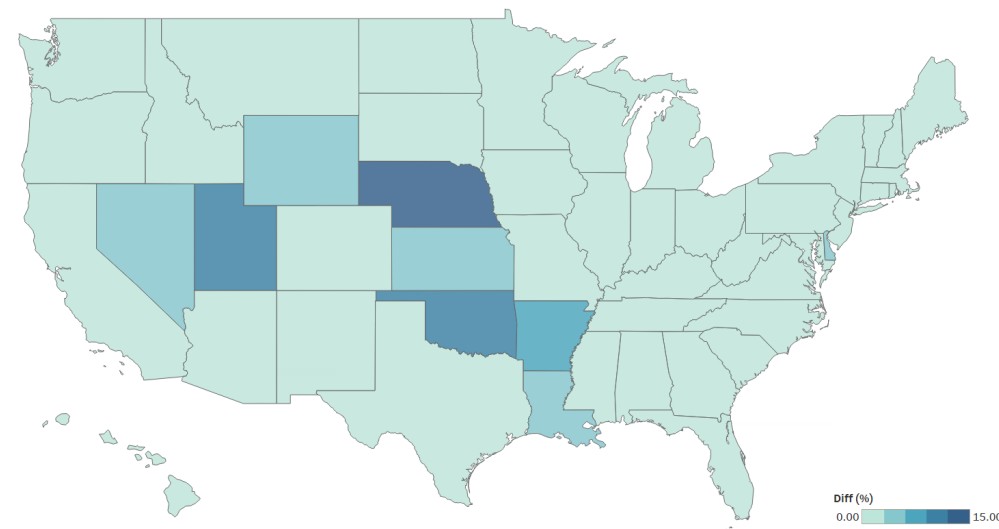

**Figure 7.** Absolute relative differences (%) between state-level survey and model point estimates for the hours per week of all hired agricultural workers.

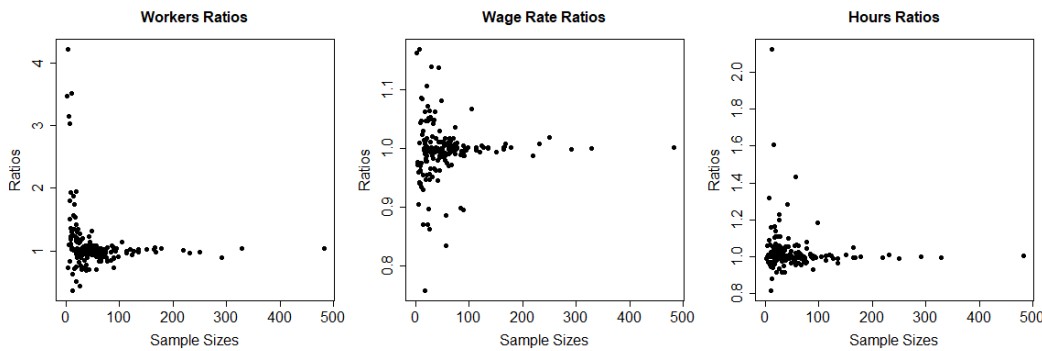

**Figure 8.** Relative measures of model estimates versus direct estimates for workers, hours, and wage rates in all four worker types.

As measured by the coefficients of variation (CVs), the state-level direct estimates were less precise than the model-based estimates for the number of workers (Figure 9), average hours worked per week (Figure 10), and average wage rates (Figure 11) for all four worker types, with an overall average reduction in the CVs of approximately 19%, 25%, and 18%, respectively.

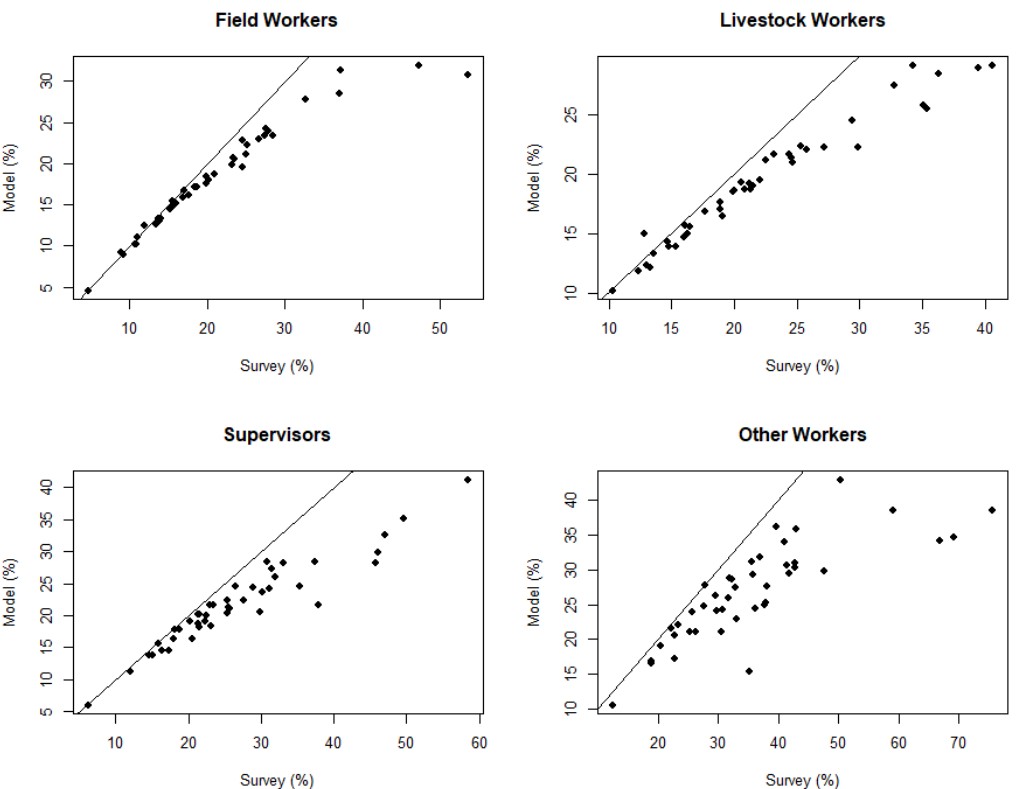

**Figure 9.** Coefficients of variation (%) for state-level model and survey estimates of number of workers by worker type.

The dramatic improvements in precision tended to be in the states with smaller sample sizes. For both quantities of interest, the "other workers" category had relatively small sample sizes. In the 44 states, the average number of reports of other workers received was 27. Two states each had just two reports of other workers. The remarkable reduction in CVs for all plotted estimates and for the "other" worker type (Figure 10) in particular is indicative of "borrowing strength" through modeling. This is often seen in small area estimation problems where the effective sample size in a small area is increased by utilizing information from larger areas, thereby increasing the precision of the estimates.

All three subarea-level models led to a reduction in the CVs of both the state and regional level estimates relative to the Farm Labor Survey direct estimates of the total number of workers, average hours per week, and average wage rates during the reference week of January 2020 (Table 1). The improvement in precision was uniform with all statistics, seeing a reduction in the model-based CVs compared with the survey CVs. The most significant improvement was in the maximums of the CVs, which had reductions approaching 50% or more.

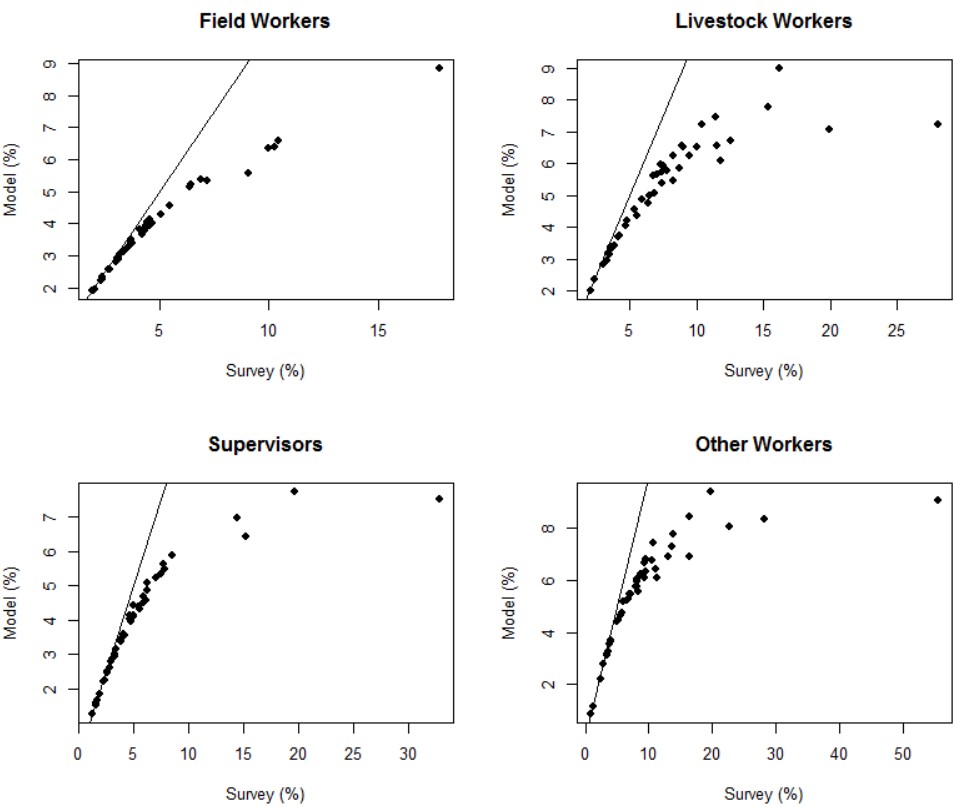

**Figure 10.** Coefficients of variation (%) for state-level model and survey estimates of average hours per week by worker type.

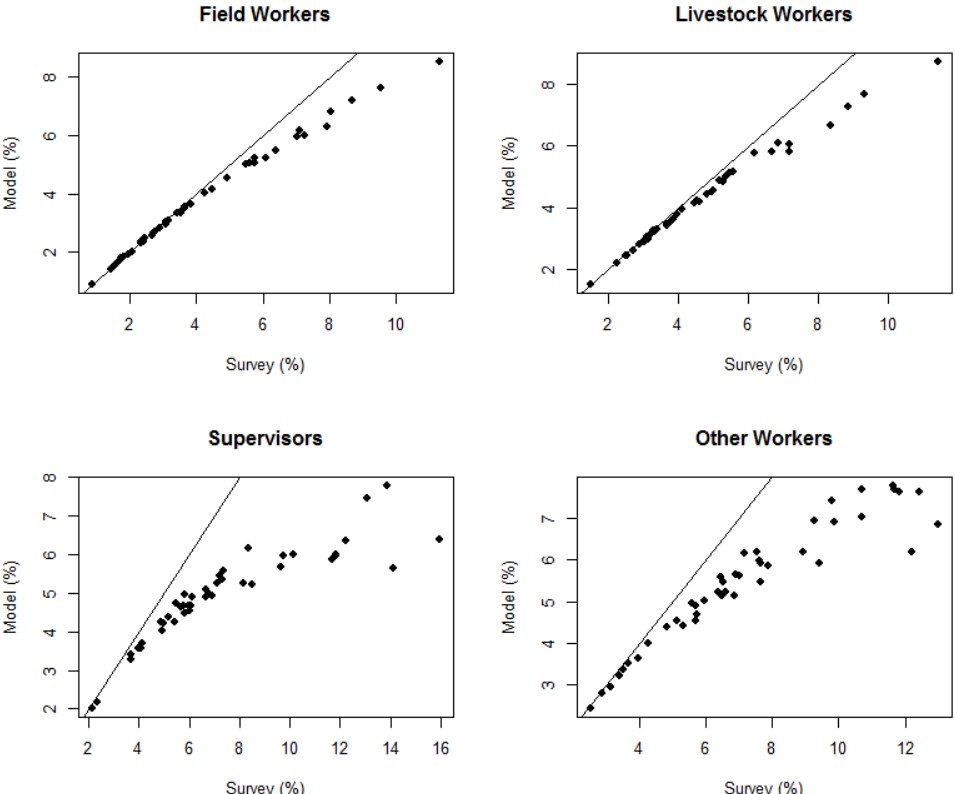

**Figure 11.** Coefficients of variation (%) for state-level model and survey estimates of average wage rates by worker type.

**Table 1.** Coefficients of variation (%) for all three estimates of hired workers at both state and regional levels.

| Quantities of Interest | | Workers | | Hours | | Wage Rate | |
|---|---|---|---|---|---|---|---|
| Level | Statistic | DE | ME | DE | ME | DE | ME |
| State | min | 3.76 | 3.65 | 1.45 | 1.35 | 1.15 | 0.99 |
| | median | 14.83 | 12.20 | 3.56 | 3.00 | 3.30 | 2.72 |
| | max | 38.72 | 22.41 | 17.72 | 5.00 | 8.03 | 4.91 |
| Region | min | 3.76 | 3.65 | 1.45 | 1.35 | 1.15 | 0.99 |
| | median | 9.10 | 7.52 | 2.71 | 2.01 | 2.03 | 1.71 |
| | max | 19.42 | 11.90 | 7.06 | 3.23 | 3.50 | 2.48 |

## 6. Conclusions

NASS has expended extensive research efforts in support of the Farm Labor program with models that provide coherent estimates at all geographic levels (state, region, and US levels) while improving the precision of estimates. In this paper, two different Bayesian subarea models were discussed: a log-normal-normal subarea model for worker totals and a normal-normal model for the hours per week and wage rate ratios. Based on a rigorous review process, NASS has been moving to adopt these small area models as the basis for official statistics in the Farm Labor program since 2020.

The contributions of this paper are threefold. First, the farm labor subarea models increased the precision of the estimates. The average CVs of the model estimates were 22% lower than the average CVs of the survey estimates for the number of workers, 18% lower for the average hours per week, and 20% lower for the average wage rates for all hired workers. The CVs were also reduced for all four worker types (see the results in Section 5).The reduction in the CVs for the state-level model estimates compared with the survey's direct estimates was a consequence of borrowing information across regions and states within regions, as well as incorporating auxiliary information such as the previous year's official estimates and other covariates. As seen in Section 5, the increase in the precision of the estimates was especially notable in states with small sample sizes.

Second, the internal benchmarking among the state level, regional level, and US level was applied at each draw from the posterior distribution instead of being applied only on the posterior means for one quantity. A practical appeal of Bayesian modeling and the choice of MCMC techniques in this application is that the state's NASS worker type domains serve as "building blocks" that produce the outputs required to support point estimates and measures of uncertainty at higher levels of aggregation, be they aggregates of geography, worker type, or even in averaging over multiple quarters in order to produce annual averages supporting the Department of Labor's regulatory obligations. Estimation of the variance of ratios (average hours and average wage rates) is also improved because all totals are treated as random rather than being known.

Third, the Farm Labor models provide measures of uncertainty for each estimate at all levels of geography. Furthermore, the methods are transparent and reproducible. Although the traditional approach of combining the available information using expert opinion resulted in accurate estimates, it did not lead to valid measures of uncertainty and lacked the transparency and reproducibility of the official statistics.

For brevity, the focus of this paper was on several estimates related to NASS worker types. Classification of worker types by SOC code is an even finer level of granularity compared with NASS worker types. However, the SOC classification can also intersect with more than one NASS worker type. For example, SOC detail pertaining to agricultural machine operators could be present in both the NASS field worker and livestock worker categories. The natural sparsity of some SOC codes nationwide, much less at the state level, suggests future lines of research related to missingness and out-of-sample prediction. The SOC class estimates are currently featured as a separate table in the Farm Labor publication, and NASS recently began publishing SOC class estimates at the regional level as well as the national level. These estimates are currently supported with simple

Bayesian Fay–Herriot models, but they share similarities with log transformations for the workers and normal-normal models for the ratios, as well as the strategies for aggregating regions to support national estimates. As each SOC code is supported by a separate set of models, future work may involve reconciling those with NASS worker type tables through benchmarking, calibration, and other adjustments.

**Author Contributions:** Conceptualization, L.C., N.B.C. and L.J.Y.; methodology, L.C. and N.B.C.; software, L.C. and N.B.C.; writing—original draft preparation, L.C. and N.B.C.; writing—review and editing, L.C., N.B.C. and L.J.Y. All authors have read and agreed to the published version of the manuscript.

**Funding:** This research was supported by the U.S. Department of Agriculture's National Agricultural Statistics Service.

**Institutional Review Board Statement:** Not applicable.

**Informed Consent Statement:** Not applicable.

**Data Availability Statement:** Not applicable.

**Acknowledgments:** The findings and conclusions in this paper are those of the authors and should not be construed to represent any official USDA or US government determination or policy. This research was supported by the U.S. Department of Agriculture's National Agricultural Statistics Service. Cruze's contributions to this research were made during his tenure at the USDA's National Agricultural Statistics Service through November 2021. The authors thank Theresa Varner and Jennifer Rhorer of the NASS for their contributions to the success of the 2020 publication. They also acknowledge Andreea Erciulescu for her contributions in the early stage of the research.

**Conflicts of Interest:** The authors declare no conflict of interest.

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
