# Peer review of "Model-Based Estimates for Farm Labor Quantities"

_stats, doi:10.3390/stats5030043_

Round 1

Reviewer 1 Report

The paper synthetically illustrates and discusses two subarea models that can be successfully used for the data coming from the USA Farm Labor Program. The variables under study are the total number of workers, according to the NASS classification, the average hours per week and the average wage rate. Two different Bayesian models are proposed and evaluated. The description of the problem, the illustration of the models and the presentation of results are satisfactory and easy to follow: the paper illustrates an interesting way to use models for answering a practical question.

In conclusion, the paper can be accepted in its present form, even if a discussion on the choice of the back-transformation for the log normal model should be necessary for a better understanding of the theory.

The Authors simply take the exponential of theta, this way obtaining the median in the original scale. Why not the mean, being this summary the most common decision in many contexts?

Some further suggestions for the Authors follow, if they wish to enrich their draft:

a-   Enriching the presentation of diagnostics and information about the shrinkage of model estimate. A stronger motivation of such diagnostics should be very reassuring for the reader. In the paper “only ARD and CV reduction are dealt.

b-    Stressing the peculiarity “…benchmarking is applied at each….”

c-       stressing that applications beyond the US rules (SOC codes, NASS worker types…) are really immediate

d-      Be careful on the fact that the 3 variables are dealt only  in Table 1, while Figures 5 and 6 deal with only two of them.

Reviewer 2 Report

Honestly, this work is not completely within my expertise therefore, my report reflects the opinion of a reader interesting in this topic and willing to learn from this paper.

In this respect, I find this paper enough interesting, well-written, and fluent in its exposition. Statistical analysis is quite elementary however the results are clearly illustrated and of certain interest to people working in this field.

Overall, I have not any objection to recommending the publication of this manuscript in its current form.

Author Response

Thank you so much for reviewing the paper.

Reviewer 3 Report

The Author focused on the hierarchical Bayesian subarea level models which are developed in support of different estimates of interest in the Farm Labor Survey. The Author introduced a case study improvement of the direct survey estimates for areas with small sample sizes by using auxiliary information and by borrowing information across areas and subareas. The resulting framework provides a complete set of coherent estimates for all required geographic levels. The article seems to be interesting for Readers. It creates a logical story and that is why in my opinion could be published after minor corrections. Some of the comments on the manuscript are listed below.

1) Line 58, 60, 77, 83, 86, 89, etc.; some sentences seem to start from small letters. Please rephrase those sentences adding some more information for example about the name of the Author you would like to cite.

2) Figure 2a; there is no numbers along abscissa. The numbers along ordinate should be rather in the form (0, 2·10-5, 4·10-5, etc.). The units of density should be also given. What is the relationship between the density used on the ordinate and the number of workers?

3) Figure 3 and 4; the numbers as well as units are missing along abscissas and ordinates.

4) Equations are not numbered and the notation used should be better described. What does the vertical line used in equation mean? If the equations are taken from the literature then the literature citation should be given.

5) The conclusions are too general and should be more precise and presented in the form of successive numeration or bullet points.
